# Synergistic Anti-Inflammatory Activity of Ginger and Turmeric Extracts in Inhibiting Lipopolysaccharide and Interferon-γ-Induced Proinflammatory Mediators

**DOI:** 10.3390/molecules27123877

**Published:** 2022-06-16

**Authors:** Xian Zhou, Sualiha Afzal, Hans Wohlmuth, Gerald Münch, David Leach, Mitchell Low, Chun Guang Li

**Affiliations:** 1NICM Health Research Institute, Western Sydney University, Westmead, NSW 2145, Australia; hans.wohlmuth@integria.com (H.W.); mitchell.low@westernsydney.edu.au (M.L.); 2Pharmacology Unit, School of Medicine, Western Sydney University, Narellan Road & Gilchrist Drive, Campbelltown, NSW 2560, Australia; s.afzal@westernsydney.edu.au (S.A.); g.muench@westernsydney.edu.au (G.M.); 3Integria Healthcare, 2728 Logan Road, Brisbane, QLD 4113, Australia; david.leach@integria.com; 4School of Chemistry & Molecular Biosciences, The University of Queensland, St. Lucia, QLD 4072, Australia

**Keywords:** ginger, turmeric, synergy, anti-inflammatory, Nrf2-HO-1, combination index

## Abstract

This study aims to investigate the combined anti-inflammatory activity of ginger and turmeric extracts. By comparing the activities of individual and combined extracts in lipopolysaccharide and interferon-γ-induced murine RAW 264.7 cells, we demonstrated that ginger-turmeric combination was optimal at a specific ratio (5:2, *w*/*w*) in inhibiting nitric oxide, tumour necrosis factor and interleukin 6 with synergistic interaction (combination index < 1). The synergistic inhibitory effect on TNF was confirmed in human monocyte THP-1 cells. Ginger-turmeric combination (5:2, *w*/*w*) also upregulated nuclear factor erythroid 2–related factor 2 activity and heme oxygenase-1 protein expression. Additionally, 6-shogaol, 8-shogaol, 10-shogaol and curcumin were the leading compounds in reducing major proinflammatory mediators and cytokines, and a simplified compound combination of 6-s, 10-s and curcumin showed the greatest potency in reducing LPS-induced NO production. Our study provides scientific evidence in support of the combined use of ginger and turmeric to alleviate inflammatory processes.

## 1. Introduction

Inflammation is the main contributing factor in the initiation and progression of many complex diseases including cardiovascular diseases [1,2], COVID-19 pneumonia [3,4,5], diabetes [6,7], cancer [8,9] and neurodegenerative diseases [10,11]. In the case of amplified and uncontrolled acute inflammation, the magnitude and the duration of the inflammatory responses are linked to the excessive production of proinflammatory modulators such as interleukin-6 (IL-6), tumour necrosis factor (TNF) and nitric oxide (NO) from activated macrophages. This may lead to harmful tissue damage, organ destruction, septic shock or even fatal consequences [12]. However, inflammation involves many proinflammatory mediators and cytokines, and a complex mechanism, which has posed great challenges for drug development [12]. Presently, steroids and nonsteroidal anti-inflammatory drugs (NSAIDs) are used as first-line anti-inflammatory medications. However, their long-term use is associated with various side effects and the efficacy of NSAIDs is limited in systemic inflammation due to their single-target behaviour on prostaglandins only rather than cytokines. Many studies have been devoted to the search for new and efficient therapeutic targets of inflammation, for example, curcumin has been considered as a promising cytokine-suppressive anti-inflammatory drug for viral pneumonia and fatal acute respiratory distress syndrome [13]. Mounting preclinical studies have demonstrated its regulatory effects on pro- and anti-inflammatory factors in systemic inflammation [13,14,15].

Combination therapy with a synergistic approach is an emerging strategy for the treatment of complex diseases such as inflammation [16]. In pharmacodynamic studies, synergy refers to an overall therapeutic effect greater than the sum of the individual effects through the positive interaction of two or more agents on the same therapeutic target or receptor [17]. In a broad view, the synergistic approach in a multi-component combination therapy offers a desired therapeutic outcome with enhanced bioactivity, and multi-target behaviour and also helps to reduce side effects or toxicity with a lower dosage required from each component [18,19,20]. For example, combination therapy in AIDs has been shown to dramatically suppress viral replication with the multi-targeted impact and reduced drug resistance [21]. Similarly, antibiotics are combined for an additional therapeutic effect against antimicrobial resistance [22]. Thus, synergy provides a more practical approach to managing multi-faceted diseases that involve a complex pathological mechanism. A rigorous mathematical analysis, the combination index (CI), has been developed and widely used to facilitate the determination and quantification of the true synergistic interactions (CI < 1) among agents in a fixed ratio [23,24,25,26].

Ginger (*Zingiber officinale* Roscoe, G) and turmeric (*Curcuma longa* L., T) are two popular functional foods belonging to the Zingiberaceae family with anti-inflammatory properties [27,28]. The observed anti-inflammatory activity of ginger is attributed to its phenolic compounds including 6-gingerol (6-g) [29] and 6-shogaol (6-s) [30] as these are the key compounds in reducing major proinflammatory mediators such as IL-6 and TNF [27]. The anti-inflammatory activity of turmeric has been extensively investigated preclinically and clinically. Its polyphenolic compounds, collectively known as curcuminoids, consist of curcumin (C) [31], demethoxycurcumin (D) and bisdemethoxycurcumin (B) [32], and are the major components responsible for the biological actions of turmeric. C is considered to be the fundamental chemical constituent contributing to the observed anti-inflammatory activity [33,34,35]. Notably, the anti-inflammatory mechanistic actions of ginger and turmeric involve many common signalling pathways and molecular targets, including Nrf2 activation [27,36,37,38].

Although the individual anti-inflammatory activity of ginger and turmeric has been investigated [27,33,34,35], the plausible synergistic activity of combined ginger and turmeric extracts has not been systematically studied. ginger and turmeric are often used in combination as popular nutraceuticals (i.e., oil, tablets, liquid, capsule); however, there is a lack of scientific evidence to support their combined use while synergy is often stated as rational. Thus, scientific evidence to support a synergistic interaction is essential to verify the claimed synergy of the combination. In addition, the interaction on associated molecular targets is also unknown. Thus, this study aims to explore the synergistic activity of extracted ginger (G) and turmeric (T) on lipopolysaccharides (LPS) and interferon-γ (IFN-γ) induced inflammation and elucidate the synergistic mechanisms by examining the active compounds’ interactions and associated molecular targets.

## 2. Results

### 2.1. Synergistic Anti-Inflammatory Effects of G-T Combinations in RAW 264.7 Cells

In the presence of LPS (50 ng/mL) and IFN-γ (50 ng/mL), a significant amount of nitrite (231.64 ± 0.01 µg/mL), IL-6 (439.64 ± 18.84 ng/mL) and TNF (1.49 ± 51.35 ng/mL) were detected from the supernatant of RAW 264.7 cells. A highly selective iNOS inhibitor, 1400 W dihydrochloride (positive control), showed a strong suppressive effect of NO with an IC_50_ value of 2.15 µg/mL.

As shown in Table 1, G exhibited moderate inhibitory effects on NO (IC_50_ = 11.78 ± 1.58 µg/mL), IL-6 (IC_50_ = 32.91 ± 9.06 µg/mL) and no effect on TNF, whereas T exhibited more potent effects on three mediators with IC_50_ = 6.51 ± 1.28, 16.10 ± 3.09 and 14.63 ± 2.19 µg/mL, respectively.

The IC_50_ values of NO inhibition by G-T combinations in the ratios of 5:5, 6:4, 7:3, 8:2, 9:1 and 5:2 (*w/w*) were lower than that of G (*p* < 0.0001) or T alone. The CI values of all tested G-T mixtures at IC_50_ were lower than 1 (excluding 1:9, 2:8 and 3:7), suggesting synergistic interactions.

In the IL-6 assay, G-T combinations at 3:7, 4:6, 5:5, 6:4 and 5:2 (*w/w*) showed lower IC_50_ values than that of G (*p* < 0.01) or T with their CI values at IC_50_ lower than 1.

In the TNF assay, only G-T 5:5 (*w/w*) showed a lower IC_50_ value than T, although all the combinations were stronger than G (*p* < 0.0001). However, CI values suggested that most of the combinations exhibited synergistic interaction in the TNF assay.

Of note, G-T combinations in the ratios of 4:6, 5:5, 6:4, 7:3, 8:2 and 5:2 constantly demonstrated synergistic interactions (CI values < 1) in reducing NO, IL-6 and TNF. Isobologram analysis (Figure 1) agreed with the synergistic activity of these G-T combinations in reducing elevated amounts of NO, IL-6 and TNF at IC_50._

We noticed that G-T 5:2 consistently showed prominent inhibitory effects on NO, IL-6 and TNF production, with IC_50_ values of 5.83 ± 0.81, 9.07 ± 1.47 and 20.07 ± 3.33 µg/mL, respectively. The IC_50_ values of G-T 5:2 were lower than that of G and T in NO and IL-6 assays. CI values of G-T 5:2 at IC_50_ in NO, IL-6 and TNF assays were determined as 0.61, 0.23 and 0.28, respectively, suggesting strong synergy.

The cytotoxicity of G and T (0–100 µg/mL) were tested on RAW 264.7 cells using an MTT assay. G exhibited insignificant cytotoxicity with an LC_50_ value estimated to be 104.3 ± 5.63 µg/mL, whereas T was moderately toxic with an LC_50_ value of 83.90 ± 7.19 µg/mL. G-T combinations (6:4, 5:2, 8:2 and 9:1) showed higher LC_50_ values (104.8 ± 6.23 to 115.80 ± 18.44 µg/mL) compared to that of G and T (*p* < 0.05) alone. Particularly, G-T 6:4 (24.53), G-T 5:2 (12.76) and G-T 5:5 (8.53) possessed the highest therapeutic index in NO, IL-6 and TNF assays, respectively. Additionally, G-T 5:2 consistently showed a higher therapeutic index than G and T individually (19.86 vs. 8.85/12.88, 12.76 vs. 3.17/5.21, 5.31 vs. 1.59/3.26) in the three assays.

### 2.2. Synergistic Anti-Inflammatory Effects of G-T Combinations in THP-1 Cells

In the presence of LPS (1 µg/mL), the amount of TNF reached 603 ± 30.64 pg/mL in the supernatant of THP-1 cells.

The TNF inhibitory effect of G (1.63–25 µg/mL) was insignificant, whereas T (1.63–25 µg/mL) showed a dose-dependent inhibition with an IC_50_ of 8.54 ± 1.43 µg/mL in THP-1 cells (Table 2 and Figure 2a).

G-T combinations (excluding 9:1) inhibited LPS-induced TNF release in a dose-dependent manner with IC_50_ ranging from 4.79 ± 1.08 to 15.58 ± 2.93 µg/mL. G-T combinations in the ratios of 1:9 to 6:4 were all significantly stronger than that of G (*p* < 0.05) and T as evidenced by lower IC_50_ values. G-T combinations (except for 9:1) also showed strong synergy (CI < 1) at IC_50_ (Figure 2a).

G (0–50 µg/mL) did not exert significant cytotoxicity in THP-1 cells, whereas T significantly reduced the cell viability at the concentrations of 25 and 50 µg/mL (*p* < 0.05) with LC_50_ value of 47.24 ± 7.84 µg/mL (Table 2). Notably, G-T combinations (4:6 to 9:1) were safer than that of T alone, with higher LC_50_ values ranging from 52.03 ± 8.97 to 118.90 ± 6.52 µg/mL. The therapeutic index of most G-T combinations (1:9 to 6:4) was higher than that of G (4.42) and T (5.53) with G-T 6:4 and demonstrated the highest therapeutic index of 10.70.

The TNF inhibitory effect of G-T 5:2 in THP-1 cells was examined to see its possible synergistic effect. As shown in Table 2 and Figure 2c, the IC_50_ value (8.32 ± 1.85 μg/mL) of G-T 5:2 was significantly lower than that of G (*p* < 0.0001) and T. G-T 5:2 exhibited a strong synergistic effect (CI = 0.49 at Fa 0.5) on TNF production (Figure 2b). G-T 5:2 was also less cytotoxic than that of T (LC_50_ value of 73.61 ± 8.05 µg/mL vs. 47.24 ± 7.84 µg/mL, *p* < 0.01), with a higher therapeutic index (8.85 vs. 5.53). The comparisons of cytotoxicity induced by G, T and G-T 5:2 in THP-1 cells are shown in Figure 2d.

### 2.3. Effect of G-T 5:2 on the Activation of Nrf2 Luciferase in AREc32 Cells

The Nrf2 up-regulatory activities of G, T and G-T combinations were tested on a luciferase assay in AREc32 cells. tBHQ (0.13–4.16 µg/mL), the positive control, increased the luminescence signal up to 11.45 ± 1.31-fold increase compared to the negative control (untreated cells). In Figure 3a, G induced the Nrf2 upregulation moderately by 2.04 ± 0.22-fold at 25 µg/mL, whereas T significantly increased the expression by 10.11 ± 1.60-fold at the same concentration. Noticeably, G-T 5:2 (25 µg/mL) boosted the Nrf2 upregulation by 18.89 ± 1.32-fold which was significantly higher than that of G (*p* < 0.001) or T (*p* < 0.001). CI-Fa curve (Figure 3b) suggested a strong synergistic Nrf2 induction by G-T 5:2. CI values were constantly lower than 1 at all tested concentrations of G-T 5:2. As shown in Figure 3c, the 24 h’s co-incubation of G-T 5:2 significantly increased HO-1 protein expression by 1.92 ± 0.37-fold (*p* < 0.05 vs. blank). The induction of HO-1 by G-T 5:2 was higher than that caused by G (1.46 ± 0.32) or T (1.03 ± 0.19) alone, although no statistical significance was detected.

### 2.4. Determining the Principal Compounds Contributing to the Synergistic Activity of G-T 5:2 Using NO Assay on RAW 264.7 Cells

In order to elucidate the role of specific compounds in the synergistic activity of G-T 5:2, major compounds from G and T were tested individually and in combination in the NO assay in RAW 264.7 cells (Table 3). Individual shogaols (6-s, 8-s and 10-s) from G demonstrated dose-dependent inhibition of NO production with IC_50_ values ranging from 1.96 ± 1.50 to 3.69 ± 0.86 µg/mL. The NO inhibitory activity of these shogaols was comparable to that of the positive control 1400 W dihydrochloride (IC_50_ = 2.15 µg/mL). By contrast, individual gingerols (6-g, 8-g and 10-g, 0.31–20 µg/mL) did not show any NO inhibition at the concentrations tested.

C (0.31–10 µg/mL) produced the strongest NO inhibition among the three tested curcuminoids from turmeric with an IC_50_ value of 5.87 ± 0.12 µg/mL, followed by D (IC_50_ = 8.30 ± 1.78 µg/mL) and B (IC_50_ = 16.14 ± 1.68 µg/mL).

To elucidate if there were positive interactions between compounds in G-T 5:2 contributing to the overall activity, different combinations of the nine active compounds were tested in the NO assay on RAW 264.7 cells (Table 3).

A mixture of three curcuminoids (C, B and D) was combined with either three gingerols (6-g, 8-g and 10-g) or three shogaols (6-s, 8-s and 10-s) in ratios equivalent to the G-T 5:2 preparation. The resulting IC_50_ values (5.52 ± 0.64 and 5.41 ± 0.70 µg/mL, respectively) were comparable to that of G-T 5:2 (5.83 ± 0.81 µg/mL) (*p* > 0.05).

Next, we made three-compound combinations composed of any two compounds from gingerols/shogaols to mix with C at ratios equivalent to their content in G-T 5:2. The IC_50_ values for two shogaols with C (ranged from 2.91 ± 0.20 to 3.82 ± 0.25 µg/mL) were significantly lower than that of G-T 5:2 (5.83 ± 0.81 µg/mL, *p* < 0.05). Particularly, 6-s, 10-s and C mixture had the lowest IC_50_ value (2.91 ± 0.20 µg/mL). However, two gingerols combined with C (IC_50_ ranged 5.12 ± 0.56–6.31 ± 0.83 µg/mL) were comparable to that of G-T 5:2 and C alone (*p* > 0.05).

When the compound combination was limited to two compounds only from G and T, respectively (using the equivalent ratios to G-T 5:2), the combinations generally showed a weak NO inhibition (IC_50_ ranged from 4.89 ± 1.07 to 43.93 ± 0.67 µg/mL) excluding 8-g-D (IC_50_ = 4.89 ± 1.07 µg/mL). CI analysis suggested that all the combinations were antagonistic in NO inhibition (CI values > 1 at IC_50,_
Appendix A).

Taken together, 6-s, 8-s and 10-s and C were the leading compounds demonstrating potent NO inhibitory activities. The three-compound combination of 6-s, 10-s and C (equivalent to their content in the G-T 5:2) showed the highest NO inhibition among all tested compounds’ combinations.

## 3. Discussion

Ginger and turmeric are two popular functional foods that are extensively used as spices, teas, dietary supplements, and natural medicines for a variety of health benefits including anti-inflammatory activity, strengthening the immune system, and relieving pain. The anti-inflammatory properties of ginger and turmeric individually have been extensively studied in preclinical and clinical studies [33,34,39,40,41]. However, little is known regarding their activity when combined in regulating inflammation which is commonly seen as nutraceuticals and complementary medicines. Our study revealed for the first time that the combined activities of G and T in certain compositions synergistically inhibited LPS and INF-γ-induced proinflammatory mediators in RAW 264.7 cells.

Synergy is defined as a combined effect of two or more agents that is larger than the sum of the effects of the individual agents [17,23]. In pharmacology, synergistic combinations can lead to improved efficacy, reduced toxicity and provide a multi-target mode of action [17]. A previous study examined the effects of a combination of ginger and turmeric powder (1:1) in reducing systematic inflammation in vivo, but potential synergistic activity was not investigated [42].

In our previous study, the G and T combination in the specific ratio of 5:2 (*w/w*) was demonstrated to exhibit synergistic activity in LPS-induced proinflammatory pathways [43]. Based on the extraction yield of the extracts used in our experiments, the G-T ratio of 5:2 (*w/w*) is equivalent to 7:10 (*w/w*) on the dried, crude rhizome basis. In the present study, we have systematically examined the combined activity of G and T in a broad range of ratios (1:9, 2:8, 3:7, 4:6, 5:5, 6:4, 7:3, 8:2, 9:1, 5:2, *w/w*) against the LPS and IFN-γ-mediated proinflammation. The demonstrated synergy in reducing major proinflammatory mediators was determined and quantified by advanced isobologram and CI models rather than just comparing IC_50_ values which could lead to false-positive records of synergy [17]. Our results demonstrated that synergy occurred in a range of ratios (3:7, 4:6, 5:5, 6:4, 7:3, 8:2 and 5:2, *w/w*). We have further demonstrated that positive interactions between G and T not only manifested as enhanced activity, but also reduced cytotoxicity in both RAW 264.7 and THP-1 cells compared with T alone. Our results showed that the IC_50_ values of G-T combinations were lower than individual ingredients at specific combination ratios, suggesting lower doses from G and T were required in the combination to reach the same level of biological activity. In addition, cell viability tests revealed that the LC_50_ values were reduced for G-T combinations (5:5, 6:4, 7:3, 8:2, 9:1, 5:2, *w/w*) in comparison to T in both RAW 264.7 and THP-1 cells, highlighting that the cytotoxicity was reduced when the dosage of T was diluted by G in the combinations. A previous study has demonstrated that C, the most abundant compound in T, significantly affected the viability of THP-1 cells when the concentration was over 40 μM [44]. Thus, the high amount of C presented in T may contribute to the observed cytotoxicity in this study. On the other hand, the reduced cytotoxicity of G-T, especially when the G is in a higher portion, is likely related to the reduced amount of C presented in the G-T combinations.

Synergistic interactions between natural products can presumably be attributed to interactions of specific chemical components. However, how the bioactive components interact in the herbal mixture that contributes to the overall activity remains to be elucidated. Although the concentration of a supposed active plant constituent may be too low to exert any clinical effect, it is a routine practice to investigate the single chemical entity as being responsible for the observed effect of the extract [19].

In our study, shogaols (6-s, 8-s and 10-s) and C were demonstrated to be principal compounds in G and T in reducing proinflammatory mediators, illustrated by the fact that their IC_50_ values were significantly lower than other tested compounds and their original extracts. Noticeably, shogaols were considerably more active and produced more potent activities than the corresponding gingerols. Shogaols and C are more lipophilic than gingerols which were found to have no significant inhibitory activity against NO. Since the G did not show the level of activity predicted based on its shogaol content, it is hypothesised that gingerols and shogaols may have opposing actions within the ginger extract. This assumption was partially confirmed in our study as shown by pair-wise testing of gingerols with shogaols that revealed antagonistic effects of all combinations except for 8g-8s and 8g-10s (Appendix A). Moreover, pairwise combinations of compounds selected from G or T were all largely found to have antagonistic and less potent activities compared with the original extract. This suggested that the combination of single active compounds from each plant does not represent the activity of the whole extract and cannot explain the synergy observed between the two extracts. A similar finding was reported for a popular herbal formula, *Salvia miltiorrhiza* Bge. and *Panax notoginseng* (Burk.) F.H.Chen, where a mixture of two principal active compounds from each herb did not exert any significant activity, whereas potent and strong synergistic effects were observed for the mixed extracts in a cell model of angiogenesis [26].

Interestingly, three curcuminoids (C, B and D) combined with three shogaols in proportions equal to G-T 5:2 showed almost identical activity in reducing NO to that of three curcuminoids combined with three gingerols. Since the IC_50_ values of both mixtures were also comparable to that of C, it may indicate that C played a major role in these compositions. Thus, we then reduced the ginger compounds to two shogaols or two gingerols and combined them with C only. The composition of two shogaols and C showed significantly lower IC_50_ values than G-T 5:2 and C alone. As the ratio was equivalent to that in the G-T 5:2 extract, it was apparent that shogaols and C exhibited a positive interaction and were likely the principal compounds contributing to the synergistic activity of G-T 5:2. Based on these results, we have formulated a simplified compound formula consisting of 6-s, 10-s and C only which demonstrated strong NO inhibition (2.91 ± 0.20 µg/mL) that was comparable to the selective NO inhibitor 1400 W dihydrochloride (IC_50_ = 2.15 µg/mL).

Nrf2 is a transcription factor that regulates cellular redox status through the endogenous antioxidant system with simultaneous anti-inflammatory activity [45,46,47]. Recent studies have revealed the pivotal role of Nrf2 in the regulation of inflammation through the Keap1 (Kelch-like ECH-associated protein)/Nrf2 (NF-E2 p45-related factor 2)/HO-1 signalling pathway which regulates anti-inflammatory gene expression and inhibits the progression of inflammation [46,48]. It has been reported that the activation of Nrf2 together with its dependent heme oxygenase (HO-1) prevents LPS-induced transcription of upregulation of proinflammatory cytokines including interleukin(IL)-6 and IL-1β [49,50]. The mechanistic action was related to orchestrating the recruitment of inflammatory cells and regulating gene expression through the antioxidant response element [51]. Thus, the Nrf2-HO-1 pathway has been considered a novel and potent therapeutic target against inflammation [52,53,54]. Our results showed that G-T (5:2) significantly upregulated the Nrf2 activity and consequently increased HO-1 protein expression, and this effect was stronger than for G or T alone. It has been well demonstrated that Nrf2-dependent HO-1 can directly suppress LPS-induced proinflammatory mediators [55] which the signalling was related to the downregulated NF-κB pathway leading to reduced proinflammation [56]. For example, sulforaphane, a natural isothiocyanate found in cruciferous vegetables, effectively inhibited LPS-stimulated proinflammatory productions of TNF, IL-1β and iNOS in primary peritoneal macrophages derived from Nrf2 (+/+) mice, whereas such effect was not seen in the Nrf2 (−/−) primary peritoneal macrophages. The expression of HO-1 was also augmented in the treated Nrf2 (+/+) macrophages, but not in Nrf2 (−/−) macrophages. Thus, it is presumed that the synergistic anti-inflammatory activities of G-T 5:2 are at least partially attributable to the upregulation of Nrf2-HO-1 axis. The anti-oxidant element related enzymes, the down-stream targets of Nrf2 and HO-1, including glutathione and glutathione S-transferases also contribute to the oxidative stress induced proinflammatory response [46]. It is the first study that linked the functional synergy to Nrf2-HO-1 signalling synergy pathway for the ginger and turmeric combination, although the contribution of Nrf2-HO-1 upregulation in the LPS-induced proinflammatory environment warrants further investigation.

In fact, previous studies have suggested that both 6-shogaol and C, the principal bioactive compounds from ginger and turmeric, exhibited anti-inflammatory and anti-oxidant activities mediated via the Nrf2-HO-1 pathway. 6-Shogaol activated Nrf2 in epithelial, HepG2, and HEK293 cells by enhancing the translocation of Nrf2 from the cytosol to the nucleus, and knockdown of Nrf2 abolished such protection, indicating that this cytoprotection is mediated by the activation of the transcription factor Nrf2 [51]. C has been repeatedly demonstrated to stimulate HO-1 gene activity by promoting the inactivation of the Nrf2-Keap1 complex, leading to increased Nrf2 binding to the resident HO-1 antioxidant response elements [57]. Such activity contributed to the protective action of C in human hepatocytes [58,59], asthmatic airway [60] and chondrocytes against inflammation [57,59]. Since shogaols and C were the key compounds contributing to the synergistic anti-inflammatory action of G and T, the interactions of shogaols and C on the activation of Nrf2 and its target genes and enzymes may provide the key to understanding the relevant mechanisms of synergy in G-T combinations. These combinations may strengthen the signalling transduction, mRNA and gene production on one or all of the pathways which resulted in stronger activities in suppressing inflammatory mediators. An in-depth investigation of shogaols and C on signalling pathways, target protein and gene expressions will be the focus of future studies. Of note, all the demonstrated synergy in extracts and compounds are based on in vitro studies, and further studies on the optimal combination with demonstrated synergy against inflammation in vivo and in clinical studies are warranted.

## 4. Materials and Methods

### 4.1. Preparation of Herbal Samples and Their Chemical Compounds

Dried ginger rhizome was provided and authenticated by Integria Healthcare (St Leonards, NSW, Australia) Pty Ltd. In excess of 100 g it was ground by an electric dry food grinder and filtered (30-mesh size). The ginger powder was mixed with 90% ethanol (1 L) and sonicated for 30 min 3 times. The solution was then filtered and dried by rotary evaporation and freeze-drying to obtain the dried extract. The turmeric powdered extract (batch J150242) was a concentrated (drug-extract ratio 25:1) ethyl acetate (99%) extract produced by Sami Labs Ltd. (Bangalore, India). As per the product specification, the extract contained C 80.2%, D 17.3% and B 2.5% as determined by high-performance liquid chromatography (HPLC). The turmeric powder was dissolved in ethyl acetate, sonicated for 30 min, filtered and dried by rotary evaporation to verify the extract was free from excipients. The dried extracts of ginger (G) and turmeric (T) were subjected to chromatographic analysis and quantification of bioactive compounds using HPLC with photodiode array detection (PDA) [61,62]. The HPLC-PDA analysis was performed on the Shimadzu UFLC system (Shimadzu, Rydalmere, NSW, Australia), comprising an LC-30AD pump, SIL-30ACHT autosampler, SPD-M20A PDA detector and DGU-20A5 inline solvent degasser. The system was controlled by Class-VP 7.4SP4 software. HPLC analysis of the extracts was performed using an Alltech Alltima (Alltech, Roseworthy, SA, Australia) reverse phase C18 column (4.6 × 150 mm I.D., 5 µm). The HPLC-PDA analysis of G and T followed the methods from [61] and the USP monograph of turmeric powder [63]. The column temperature was kept at 30 °C throughout the analysis. The samples were kept at 4 °C. The injection volume was 25 µL. The PDA (UV 200–500 nm) was recorded, and UV 280 nm was used to quantify the marker chemical compounds listed below. The method validation parameters and the HPLC chromatograms are shown in Appendix A.

G and T were re-dissolved in dimethyl sulfoxide (DMSO, Sigma Aldrich, Castle Hill, NSW, Australia) and subjected to the cellular bioassays. The G-T combinations were prepared by mixing the same concentration (50 mg/mL) of G and T in ten different ratios (1:9, 2:8, 3:7, 4:6, 5:5, 6:4, 7:3, 8:2, 9:1, and 5:2, *w/w*).

Chemical standards of 6-gingerol (6-g), 8-gingerol (8-g), 10-gingerol (10-g), 6-shogaol (6-s), 8-shogaol (8-s), 10-shogaol (10-s) from ginger, and C, D and B from turmeric were purchased from Chengdu Biopurify Phytochemicals Ltd. (Chengdu, China; purity > 98%). Their identity and purity were confirmed by HPLC-PDA. The stock solutions of these reference compounds were prepared in DMSO for the bioassays or stored at −20 °C until use. The calculated amount of each bioactive in G, T and G-T 5:2 extracts (mg/mL) is shown in Appendix A.

### 4.2. Cell Culture

The murine RAW 264.7 macrophages [24] and human mammary MCF7-derived reporter cell line AREc32 (AREc32) were cultured at 37 °C in Dulbecco’s Modified Eagle Medium (DMEM) (Lonza, Norwest, NSW, Australia) supplemented with 10% foetal bovine serum (FBS) (Life Technologies, Scoresby, VIC, Australia), and 1% penicillin-streptomycin (Life Technologies, Australia) in a humidified atmosphere containing 5% CO_2_.

Human monocytic THP-1 cells, derived from an acute monocytic leukaemia (ATCC TIB-202, Rockville, MD, USA), were maintained in Roswell Park Memorial Institute Medium 1640 culture medium (Lonza, Australia) containing 10% FBS and 1% penicillin-streptomycin (Thermo Fisher Scientific, Scoresby, VIC, Australia). THP-1 cells monocytes were differentiated into macrophages after the incubation with 100 nM phorbol 12-myristate 13-acetate (PMA, Sigma, Point Cook, VIC, Australia) for 24 h.

### 4.3. Nitric Oxide Assay

The NO production in RAW 264.7 cells was measured by its stable metabolite nitrite using the Griess reaction [24]. LPS from Escherichia coli 0111:B4 purified by trichloroacetic acid extraction (Sigma, Australia) and murine recombinant IFN-γ (Lonza, Australia) were used to stimulate NO production. Briefly, RAW 264.7 cells (density at 1 × 10^6^/mL) were seeded on a 96-well cell culture plate (Corning Costar, Sigma, Australia) and incubated for 48 h followed by pre-treatments with individual and combined extracts/compounds in 0.1% DMSO. After incubation for 2 h, LPS (50 ng/mL) and IFN-γ (50 ng/mL) were added to the cells and co-incubated for another 18 h. After the stimulation, 100 µL of cells supernatant was collected and mixed with Griess reagent (1% sulfanilamide in 5% phosphoric acid and 0.1% N-1-naphthylethylenediamine dihydrochloride in Milli-Q water) for NO measurement at 540 nm using a microplate reader (BMG LABTECH, FLUOstar OPTIMA, Mount Eliza, VIC, Australia).

### 4.4. TNF and IL-6 ELISA Assay

The cell supernatants from RAW 264.7 cells treated with various samples were collected 24 h after the stimulation with LPS and IFN-γ. The cell supernatants were analysed for TNF and IL-6 using commercial ELISA kits (Lonza, Australia) according to the manufacturer’s instructions.

THP-1 cells primed with phorbol myristate acetate (Sigma, Australia) were co-incubated with samples for 2 h before stimulation with LPS (1 µg/mL) for 24 h to enable the measurement of TNF using commercial ELISA kits (Lonza, Australia).

### 4.5. Cell Viability Assays

The cytotoxicity of RAW 264.7 and THP-1 were determined using the methyl thiazolyl tetrazolium (MTT) and alamar blue assay, respectively. After the media was withdrawn from cells, MTT (0.12 mg/mL) or alamar blue (10 µg/mL) in phosphate-buffered saline were added to the cells and incubated for 4 h. For the MTT assay, the supernatant was then discarded and replaced with 100 µL DMSO (Sigma, Australia) and the optical density was measured using a microplate reader (BMG LABTECH, FLUOstar OPTIMA, Australia) at 510 nm. For the alamar blue assay, the fluorescent absorbance of alamar blue was measured at 540 nm excitation and 590 emissions using the microplate reader. The absorbance in control (medium with 0.1% DMSO) cells was taken as 100% of cell viability [24].

### 4.6. Determination of Nrf2 Expression by Luciferase Assay

The luciferase assay quantifying total Nrf2 protein in cells was conducted as previously reported, with modifications [64,65]. MCF-7 AREc32 cells (transfected with Nrf2) were cultured and seeded at a density of 1.0 × 10^6^ cells/mL in 96-well plates. After 48 h incubation, the cells were activated with tert-butylhydroquinone (tBHQ) as a positive control, individual or combined extracts of G and T, or medium with 0.1% DMSO (negative control). Cells were then incubated with alamar blue (10 µg/mL resazurin) to examine the cell viability. After reading the plates with excitation at 530 nm and emission at 590 nm, the alamar blue solution was replaced with 20 µL of triton lysis buffer (tris HCl: 1.705%, tris base: 0.508%, 5M NaCl: 1.5%, 1M MgCl2: 0.3%, Triton X 100 pure liquid: 0.75%) for 20 min at −20 °C. Then the cell lysates (15 µL) were transferred to a fresh 96-well plate and mixed with 100 µL of luciferin buffer (D-luciferin 30 mg/mL: 0.525%, DTT 1M: 3%, coenzyme A 10 mM: 1.5%, ATP 100 mM: 0.45%) for 30 min. The fluorescence was measured within 30 min at an excitation wavelength of 488 nm and an emission wavelength of 525 nm. The activation of Nrf2 was calculated by the fold change compared to that of the negative control (cells with medium only).

### 4.7. Determination of HO-1 Protein Expression by Western Blotting Analysis

RAW 264.7 cells were incubated with media, G, T or G-T (5:2, *w/w*) at 12.5 µg/mL in T75 cell flasks for 24 h before the stimulation of LPS (1 µg/mL). The cells were then lysed with RIPA lysis buffer (Thermo Fisher Scientific, Australia) and the protein was collected and quantified using a BCA quantification kit (Thermo Fisher Scientific, Australia). An equal amount of protein from each sample was then separated by SDS-PAGE and transferred to a PVDF membrane. The membrane was blocked by 5% milk, then incubated overnight at 4 °C with rabbit polyclonal antibodies against HO-1 (1:1000) (cell signalling technologies, Danvers, MA, USA), or β-actin (1:1000) (Cell Signalling Technologies, USA), followed by horseradish peroxidase-conjugated secondary antibodies. Then the membranes were exposed with Pierce ECL Plus Western blot substrate (Thermo Fisher Scientific, Australia). The band intensities of the membranes were quantified by ImageJ, and the control for equivalent protein loading was assessed by anti-β-actin antibody.

### 4.8. Determination of Synergistic, Additive or Antagonistic Interactions

Potential interactions between G and T and their compounds were determined using the CompuSyn software 2.0 (Biosoft, San Francisco, CA, USA) based on the Chou-Talalay method [66]. The concentration-response curves of the individual and combined extracts/compounds pertaining to the bioassays were constructed and entered into the program. The relevant statistics, combination index-fraction affected (CI-Fa) curves and isobologram figures were then generated. The CI values were used to demonstrate interaction as follows: synergism (CI < 1), additive effect (CI = 1), and antagonism (CI > 1) [17]. The CI-Fa curve suggested the correlation between the CI values and the suppressive effect (i.e., on NO) [25,26].

### 4.9. Statistical Analysis

All data were expressed as mean ± STD (*n* ≥ 3) and analysed by GraphPad Prism 8 (Dotmatics, Boston, MA, USA). The therapeutic index was calculated as the LC_50_ value (median lethal concentration) divided by the IC_50_ value (median inhibitory concentration). The difference between groups was analysed by one-way ANNOVA. Values of *p* < 0.05 were considered statistically significant.

## 5. Conclusions

Extracts of ginger and turmeric were combined in ratios ranging from 4:6 to 8:2 (*w/w*) synergistically reduced LPS-induced NO, TNF, and IL-6 expression in RAW 264.7 cells, as well as TNF in THP-1 cells. Interestingly, the combination also displayed higher therapeutic indexes. In particular, the optimal combination of G-T extracts at the ratio of 5:2, *w/w* [equivalent to 7:10 (*w/w*) for the starting material] further enhanced Nrf2 and HO-1 protein expressions, which may contribute to the anti-inflammatory activity. Shogaols (6-s, 8-s, 10-s) and curcumin were the leading compounds in reducing major proinflammatory mediators and cytokines, and a simplified compound combination of 6-s, 10-s and curcumin showed the greatest potency in reducing LPS-induced NO production. In summary, this study provides evidence at molecular levels to support the combined use of ginger and turmeric with a synergistic approach to reduce proinflammatory mediators which the mechanism was at least partially related to upregulated Nrf2 activation and interactions among the key bioactive compounds.

## 6. Patents

The authors declare that the research was conducted in the absence of any commercial or financial relationships that could be construed as a potential conflict of interest. DL and HW are employees of Integria Healthcare (Australia) Pty Ltd. which provided funding and in-kind support for the work as an Australian Research Council Linkage Project industry partner. A patent application (Australian Mechanism of synergy in ginger and turmeric. Patent Application No. 2021902926) based in part on these results has been filed by Integria Healthcare (Australia) Pty Ltd. naming XZ, GM, ML, DL HW, CGL and others as inventors.

## Figures and Tables

**Figure 1 molecules-27-03877-f001:**
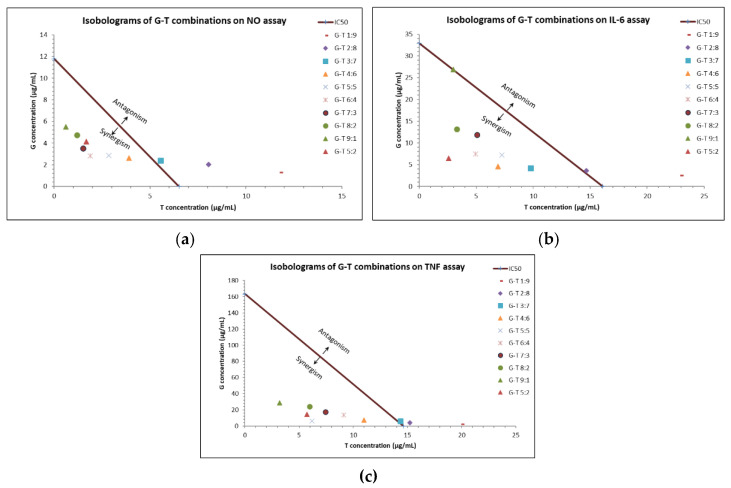
Interactions of G-T combinations (1:9, 2:8, 3:7, 4:6, 5:5, 6:4, 7:3, 8:2, 9:1, 5:2, *w/w*) in reducing LPS-induced NO, IL-6 and TNF in RAW 264.7 cells as analysed by isobologram. Isobologram of G-T combinations at IC50 on NO (**a**), IL-6 (**b**) and TNF (**c**) assays.

**Figure 2 molecules-27-03877-f002:**
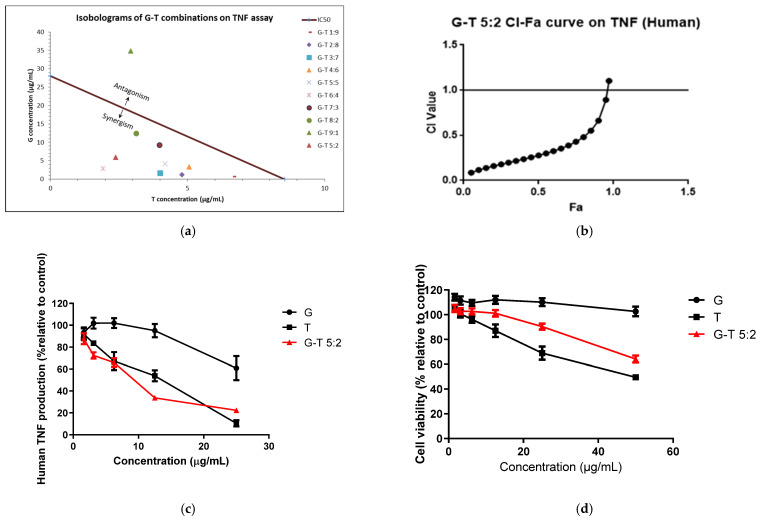
Synergistic interaction of G-T combination s (1:9, 2:8, 3:7, 4:6, 5:5, 6:4, 7:3, 8:2, 9:1, 5:2, *w/w*) in in-hibiting LPS-induced TNF in THP-1 cells. Isobologram of G-T combinations in reducing TNF at IC50 (**a**), CI-Fa curve of G-T 5:2 (**b**), dose-response curves of G, T and G-T 5:2 on TNF assay (**c**) and cell viability (**d**) in THP-1 cells (*n* ≥ 3). The IC50 values of G and G-T 9:1 used in the isobologram (**a**) were estimated from the dose-response curves which were at 28.04 ± 3.99 and 38.70 ± 8.66 µg/mL, re-spectively.

**Figure 3 molecules-27-03877-f003:**
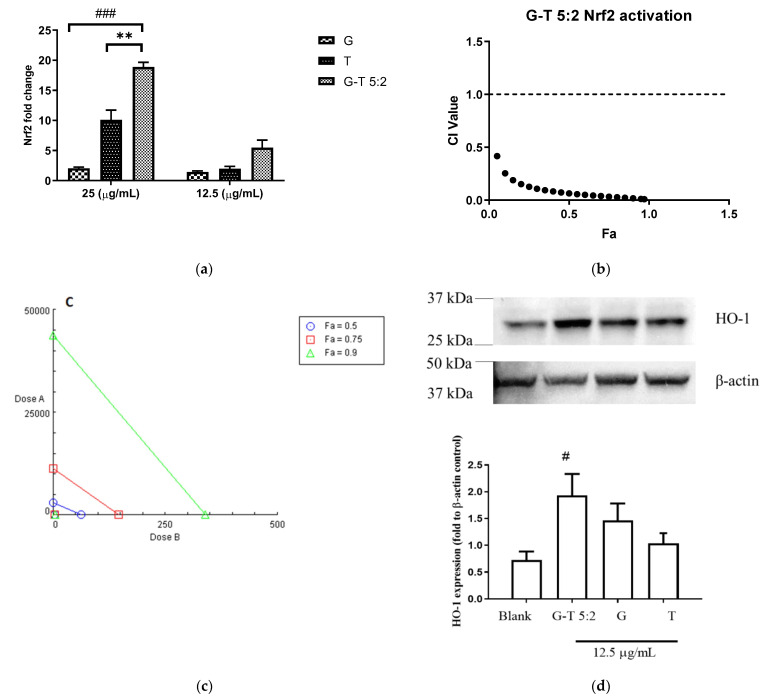
Up-regulatory effect of G, T and G-T 5:2 on Nrf2 and HO-1 protein levels (*n* ≥ 3). (**a**) G, T and G-T 5:2 induced Nrf2 luciferase at 25 and 12.5 µg/mL (*n* ≥ 3). ** *p* < 0.01 compared between G-T 5:2 and T at 25 µg/mL; ^###^
*p* < 0.001 compared between G-T 5:2 and G at 25 µg/mL. (**b**) CI-Fa curve of G-T 5:2 in upregulating Nrf2 expression and strong synergy was observed at all tested concentration levels. (**c**) Isobologram of Nrf2 luciferase by G, T and G-T 5:2. (**d**) G-T 5:2 further strengthened the en-hancement of HO-1 protein expression as analysed by western blot analysis (*n* = 3). ^#^
*p* < 0.05 compared between G-T 5:2 and blank.

**Table 1 molecules-27-03877-t001:** Anti-inflammatory activity, synergistic interaction, therapeutic index and cytotoxicity parameters of ginger extract (G), turmeric extract (T) and G-T combinations in inhibiting LPS-induced NO, IL-6 and TNF productions in RAW 264.7 cells (*n* > 3 of experiments).

Extracts and Combinations	Cell Viability LC_50_ (µg/mL) (Mean ± STD)	NO			IL-6			TNF		
		IC_50_ (µg/mL) (Mean ± STD)	CI Value at IC_50_	Therapeutic Index	IC_50_ (µg/mL) (Mean ± STD)	CI Value at IC_50_	Therapeutic Index	IC_50_ (µg/mL) (Mean ± STD)	CI Value at IC_50_	Therapeutic Index
G	104.3 ± 5.63 *	11.78 ± 1.58	N/A	8.85	32.91 ± 9.06	N/A	3.17	163.40 ± 3.94 *	N/A	0.64
T	83.90 ± 7.19	6.51 ± 1.28	N/A	12.88	16.10 ± 3.09	N/A	5.21	14.63 ± 2.19	N/A	5.73
G-T 1:9	72.29 ± 4.77 ^∆^	13.08 ± 0.97	1.91	5.52	25.46 ± 4.28	1.50	2.84	22.20 ± 6.13 ^∆^	1.86	3.26
G-T 2:8	76.93 ± 6.72 ^∆^	10.08 ± 0.96	1.41	7.63	18.33 ± 3.68 ^∆^	1.01	4.20	19.02 ± 3.09 ^∆^	1.10	4.04
G-T 3:7	82.15 ± 3.00 ^∆^	7.93 ± 0.81 ^∆^	1.05	10.36	13.98 ± 2.00 ^∆^	0.70	5.88	20.51 ± 5.17 ^∆^	0.40	4.01
G-T 4:6	81.00 ± 2.86 ^∆^	6.52 ± 0.70 ^∆^	0.83	12.42	11.52 ± 2.33 ^∆^	0.52	7.03	18.30 ± 2.64 ^∆^	0.73	4.43
G-T 5:5	102.70 ± 2.10	5.72 ± 0.62 ^∆^	0.69	17.95	14.47 ± 3.04 ^∆^	0.59	7.10	12.40 ± 2.16 ^∆^	0.31	8.53
G-T 6:4	115.80 ± 18.44 ^&^	4.72 ± 0.48 ^∆^	0.54	24.53	12.40 ± 2.02 ^∆^	0.45	9.34	22.81 ± 5.38 ^∆^	0.57	5.08
G-T 7:3	93.29 ± 5.04	5.02 ± 0.23 ^∆^	0.53	18.58	16.94 ± 2.36 ^∆^	0.53	5.51	24.79 ± 4.58 ^∆^	0.62	3.76
G-T 5:2	115.80 ± 11.4 ^&^	5.83 ± 0.81 ^∆^	0.61	19.86	9.07 ± 1.47 ^∆^	0.23	12.76	20.07 ± 3.33 ^∆^	0.28	5.77
G-T 8:2	104.80 ± 6.23 ^&^	5.92 ± 1.18 ^∆^	0.59	17.70	16.53 ± 3.79 ^∆^	0.43	6.34	29.92 ± 6.24 ^∆^	0.49	3.50
G-T 9:1	107.40 ± 8.67 ^&^	6.11 ± 0.96 ^∆^	0.57	17.58	29.87 ± 7.66	1.00	3.60	31.91 ± 6.16 ^∆^	0.41	3.37

^∆^ *p* < 0.05 compared with G; ^&^ *p* < 0.05 compared with T as analysed by one-way ANOVA test. * estimated LC_50_ or IC_50_ value based on the trend of the dose-response curve.

**Table 2 molecules-27-03877-t002:** Anti-inflammatory, synergistic interaction, therapeutic index and cytotoxicity parameters of ginger and turmeric extracts in inhibiting LPS-induced TNF production in THP-1 cells (*n* > 3 of experiments).

Extracts and Combinations	Cell Viability LC_50_ (µg/mL)	IC_50_ (µg/mL) Mean ± STD	CI Value at IC_50_	Therapeutic Index
G	123.8 ± 9.10	28.04 ± 3.99	N/A	N/A
T	47.24 ± 7.84	8.54 ± 1.43	N/A	5.53
G-T 1:9	40.94 ± 11.08 ^∆^	7.40 ± 1.32 ^∆^	1.52	5.53
G-T 2:8	41.16 ± 10.90 ^∆^	5.99 ± 0.70 ^∆^	0.60	6.87
G-T 3:7	45.30 ± 6.74 ^∆^	5.72 ± 0.57 ^∆^	0.11	7.92
G-T 4:6	52.03 ± 8.97 ^∆^	8.43 ± 1.25 ^∆^	0.71	6.17
G-T 5:5	53.16 ± 7.79 ^∆^	8.37 ± 1.44 ^∆^	0.64	6.35
G-T 6:4	51.26 ± 7.62 ^∆^	4.79 ± 1.08 ^∆^	0.33	10.70
G-T 7:3	52.61 ± 5.41 ^∆^	13.24 ± 3.11 ^∆^	0.80	3.97
G-T 5:2	73.61 ± 8.05 ^∆,&^	8.32 ± 1.85 ^∆^	0.49	8.85
G-T 8:2	56.35 ± 6.25 ^∆^	15.58 ± 2.93 ^∆^	0.81	3.62
G-T 9:1	118.90 ± 6.52 ^&^	N/A	N/A	N/A

^∆^*p* < 0.05 compared with G; ^&^
*p* < 0.05 compared with T as analysed by one-way ANNOVA test. N/A: not available.

**Table 3 molecules-27-03877-t003:** Inhibitory effects of individual and combined bioactives of ginger (G) and turmeric (T) in inhibiting LPS-induced NO expression in RAW 264.7 cells (*n* > 3).

Bioactive Compounds in G and T	Content in G or T (mg/g)	IC_50_ (µg/mL) of Individual Compounds (µg/mL)	IC_50_ (µg/mL) of Paired Combinations Equivalent to Their Ratio in G-T 5:2				IC_50_ (µg/mL) of Three-Compound Combinations Equivalent to Their Ratio in G-T 5:2		IC_50_ (µg/mL) of Six-Compound Combinations Equivalent to Their Ratio in G-T 5:2	
			Compounds in G	C	B	D	Compounds in G	C	Compounds in G	C, B, D
6-g	69.57 ± 0.16	ND *	6-g	17.21 ± 2.41	43.93 ± 0.67	9.44 ± 1.42	6-g, 8-g	6.31 ± 0.83	6-g, 8-g, 10-g	5.52 ± 0.64
8-g	10.43 ± 0.23	ND *	8-g	12.31 ± 4.41	27.33 ± 3.75	4.89 ± 1.07	6-g, 10-g	5.37 ± 0.58	6-s, 8-s, 10-s	5.41 ± 0.70
10-g	19.62 ± 0.63	ND *	10-g	10.02 ± 5.28	17.34 ± 5.38	11.86 ± 2.86	8-g, 10-g	5.12 ± 0.56	**Extracts**	**IC_50_ (μg/mL)**
6-s	7.48 ± 0.19	2.90 ± 1.22	6-s	16.06 ± 1.12	12.70 ± 2.09	9.06 ± 1.16	6-s, 8-s	3.34 ± 0.44	G	11.78 ± 1.58
8-s	1.56 ± 0.03	1.96 ± 1.50	8-s	13.19 ± 1.91	12.61 ± 1.64	6.33 ± 0.81	6-s, 10-s	2.91 ± 0.20	T	6.51 ± 1.28
10-s	2.30 ± 0.03	3.69 ± 0.86	10-s	10.02 ± 1.95	40.32 ± 2.50	9.68 ± 0.92	8-s, 10-s	3.82 ± 0.25	G-T 5:2	5.83 ± 0.81
C	751.76 ± 101.45	5.87 ± 0.12								
B	14.77 ± 3.63	16.14 ± 1.68								
D	156.15 ± 26.24	8.30 ± 1.78								

* ND: Inhibition not detected.

## Data Availability

The data presented in this study are available on request from the corresponding author. The data are not publicly available due to the full RCT application related to the patent application (Australian Mechanism of synergy in ginger and turmeric. Patent Application No. 2021902926).

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
