# Peer review of "Synergistic Anti-Inflammatory Activity of Ginger and Turmeric Extracts in Inhibiting Lipopolysaccharide and Interferon-γ-Induced Proinflammatory Mediators"

_molecules, 2022, doi:10.3390/molecules27123877_

Round 1
Reviewer 1 Report
The article by Zhou et al describes the anti-inflammatory properties of the combination ginger and turmeric suggesting the activation of nrf2 HO-1 axis. Some concerns are requested:
- The HO-1 expression was measured only in basal conditions and not in presence of LPS which mimics in vitro the pro-inflammatory enviroment. In order to state the HO-1 in the title, more data on enzymatic activity, siRNA HO-1 or HO-1 inhibitor should support the protein involvement.
- Table 2 is not mentioned for MTT results discussion.
- Curcumin is a strong HO-1 inducer but no references were included
Author Response
Dear Reviewer,
Many thanks for the comments. Please see our response below per your comment:
-The HO-1 expression was measured only in basal conditions and not in presence of LPS which mimics in vitro the pro-inflammatory enviroment. In order to state the HO-1 in the title, more data on enzymatic activity, siRNA HO-1 or HO-1 inhibitor should support the protein involvement.
Response: We have modified the title to remove the “regulation of Nrf2-HO-1”. We have also elaborated the discussion to strengthen the link between the upregulated HO-1 and reduced proinflammatory response based on the previous publications. In general, the effect of HO-1 inducer in the pro-inflammatory environment is investigated in animals, thus we may consider conducting the enzymatic activity, siRNA HO-1 or HO-1 inhibitor using appropriate animal models in our future study.
- Table 2 is not mentioned for MTT results discussion.
Response: We have cited Table 2 in line 159 for the MTT results. We have also elaborated the discussion on the MTT results of Table 2 on lines 266-271.
- Curcumin is a strong HO-1 inducer but no references were included
Response: We have elaborated the discussion of curcumin as a strong HO-1 inducer in lines 343-347. Please noted that curcumin is abbreviated as “C” in our manuscript.
Thanks and regards,
Xian on behalf of all authors
Reviewer 2 Report
The manuscript entitled “Synergistic anti-inflammatory activity and regulation of Nrf2 and HO-1 by ginger and turmeric” represent interesting data and on my opinion it falls within the scope of Molecules. This paper presents an interesting and carefully designed research. However, manuscript should be improved. In order to make the work more interesting and understandable, it seems us to be necessary to make some minor modifications:
- The manuscript is very long and difficult to read and the results section should be shortened and in my opinion could be easily shortened by an half of its length without changing the general sense of the article.
- The authors should describe how the therapeutic index was determined.
- Please provide more details on LC-MS and HPLC-PDA analyses. The authors only mention that ginger and turmeric extracts and their bioactive substances were analyzed by liquid chromatography-mass spectrometry (LC-MS) and HPLC-PDA, but do not provide details.
Author Response
Dear Reviewer,
Many thanks for your comments. We have provided the responses below as per your comment.
- The manuscript is very long and difficult to read and the results section should be shortened and in my opinion could be easily shortened by an half of its length without changing the general sense of the article.
Response: We have shortened the result section for almost half as suggested.
- The authors should describe how the therapeutic index was determined.
Response: We have included the calculation of therapeutic index in the section of “Statistical analysis”, lines 478-479.
- Please provide more details on LC-MS and HPLC-PDA analyses. The authors only mention that ginger and turmeric extracts and their bioactive substances were analyzed by liquid chromatography-mass spectrometry (LC-MS) and HPLC-PDA, but do not provide details.
Response: Since we only included the data (chromatogram, validation and quantification results) of HPLC-PDA in the supplementary material, we have removed LC-MS and elaborated the details of HPLC-PDA in the section of “Materials of Methods”, lines 373-383.
Yours sincerely,
Xian Zhou on behalf of all authors
Round 2
Reviewer 1 Report
The authors improved the manuscript according reviewer's suggestion